# Melatonin as the Cornerstone of Neuroimmunoendocrinology

**DOI:** 10.3390/ijms23031835

**Published:** 2022-02-06

**Authors:** Igor Kvetnoy, Dmitry Ivanov, Ekaterina Mironova, Inna Evsyukova, Ruslan Nasyrov, Tatiana Kvetnaia, Victoria Polyakova

**Affiliations:** 1Center of Molecular Biomedicine, Saint-Petersburg Research Institute of Phthisiopulmonology, 191036 Saint-Petersburg, Russia; igor.kvetnoy@yandex.ru; 2Department of Physiology and Department of Pathology, Saint-Petersburg State University, 199034 Saint-Petersburg, Russia; 3Department of Pathology, Saint-Petersburg State Pediatric Medical University, 194100 Saint-Petersburg, Russia; doivanov@yandex.ru (D.I.); rrmd99@mail.ru (R.N.); vopol@yandex.ru (V.P.); 4Department of Biogerontology, Saint Petersburg Institute of Bioregulation and Gerontology, 197110 Saint-Petersburg, Russia; kvetnaia@gerontology.ru; 5Department of Perinatal Pathology, Ott Research Institute of Obstetrics, Gynecology and Reproductology, 199034 Saint-Petersburg, Russia; eevs@yandex.ru

**Keywords:** melatonin, neuroimmunoendocrinology, homeostasis, antioxidant activity, anti-inflammatory function

## Abstract

Much attention has been recently drawn to studying melatonin – a hormone whose synthesis was first found in the epiphysis (pineal gland). This interest can be due to discovering the role of melatonin in numerous physiological processes. It was the discovery of melatonin synthesis in endocrine organs (pineal gland), neural structures (Purkinje cells in the cerebellum, retinal photoreceptors), and immunocompetent cells (T lymphocytes, NK cells, mast cells) that triggered the evolution of new approaches to the unifield signal regulation of homeostasis, which, at the turn of the 21st century, lead to the creation of a new integral biomedical discipline — neuroimmunoendocrinology. While numerous hormones have been verified over the last decade outside the “classical” locations of their formation, melatonin occupies an exclusive position with regard to the diversity of locations where it is synthesized and secreted. This review provides an overview and discussion of the major data regarding the role of melatonin in various physiological and pathological processes, which affords grounds for considering melatonin as the “cornerstone” on which neuroimmunoendocrinology has been built as an integral concept of homeostasis regulation.

## 1. Introduction

In the recent decade, particular research attention has been attracted to a wide range of melatonin’s biological activities and its role in cellular activity, regulation of intercellular and intersystem relationships, which provides consistency of the body’s internal environment and its protection during interaction with the changing external environment [1,2,3,4,5].

Melatonin (MT) is produced not only in the pineal gland but also in the gastrointestinal tract, brain, liver, kidney, adrenal gland, heart, thymus, genital glands, placenta, uterus, platelets, eosinophilic leukocytes, natural killer cells and other immune system cells [6,7,8,9,10] (Figure 1).

MT is synthesized from the tryptophan amino acid which, through hydroxylation and decarboxylation, turns into serotonin, from which MT is produced with the help of N-acetyltransferase (NAT) and hydroxyindole O-methyltransferase (HIOMT) enzymes. The endocrine function of the pineal gland is controlled by hypothalamic suprachiasmatic nuclei (HSN) and has a circadian rhythm.

The photic information from retinal ganglion cells passes through the retino-hypothalamic tract to the suprachiasmatic nucleus (SCN), from which signals go to the superior cervical ganglia and then reach the pineal gland via sympathetic noradrenergic nerves and activate pinealocytes. Light suppresses the MT production and secretion, and therefore its maximum level in the pineal gland and human blood is observed at midnight and the minimum during the daytime [11,12].

Pineal MT is released into the blood and cerebrospinal fluid, while MT, synthesized in the cells of the diffuse neuroimmunoendocrine system [7,13] gets to the blood in small amounts and produces the paracrine and autocrine effects in the places of its syn- thesis. Being hydrophilic, the MT molecule is highly lipophilic and therefore can easily pass through the blood-brain barrier, reaching capillaries where 70% of MT binds to albumin. MT is metabolized not only in the liver but also in other tissues (brain, intestine) where specialized enzymes have been found [14,15]. MT has a regulating effect through binding to receptors. In humans, two types of membrane receptors (MT1 and MT2) have been identified along with their localization in chromosomes (4q35 and 11q21-22), as well as nuclear receptors (RORα/RZR) [9,16] and a lower-affinity cytosolic binding site, designated MT3. MT3 has recently been identified as QR2 (quinone reductase 2) which is of significance since it links the antioxidant effects of melatonin to a mechanism of action [17]. The results show the existence of several putative cytosolic melatonin receptors including enzyme quinone reductase 2 (the MT3 receptor), RORα/RZR nuclear receptors, and calmodulin. Evidence points to immune cells as the main sites of RORα1 and RORα2 responsible for mediating melatonin effects while RZRβ is abundant in the pineal gland. RORα was shown to be an active receptor in regulating of antioxidant enzymes controling of inflammation and oxidative stress [18]. MT receptors have been found in hypothalamic suprachiasmatic nuclei, cortex cerebellum, retina, spleen, liver, genital and mammary glands, uterus, thymus, gastrointestinal tract, thrombocytes and lymphocytes [19]. In the brain, numerous membrane protein receptors of MT have been found which are paired with the guanine-nucleotide-binding protein (G protein) and to the maximum extent represented in the hypothalamus and pineal gland [16,20,21].

MT can pass through the membrane, bind to protein receptors on the nucleus surface, penetrate the nucleus and realize its action at the level of nuclear chromatin, with a direct effect on the protein synthesis by the cell’s genetic apparatus. Having a high permeability even in the absence of receptors, the MT molecule has a systemic effect at the cellular level by modulating the cytoskeleton and mitotic function through binding to calmodulin and as a free oxygen radical scavenger [3].

## 2. Pineal Melatonin

Melatonin (N-acetyl-5-methoxytryptamine) was discovered and isolated from bovine pineal gland extracts by American dermatologist Aaron Lerner in 1958. Lerner, a researcher at Yale University studying the nature of vitiligo, tried to find chemical substances responsible for skin pigmentation. He found an article dated 1917 [22], which reported that feeding extract of the pineal glands of cows when placed in a jar with tadpoles lightened their skin within 30 min. The tadpole skin became so transparent that one could observe the workings of their heart and intestine.

In 1958, Lerner isolated an extract from bovine pineal glands that lightened frog skin. Then all the research was directed at searching for a key component. Lerner understood that he was looking for a hyperactive hormone capable of decolorizing the skin thousands of times more effectively than adrenaline. Lerner and colleagues processed 250,000 pineal glands, yet extracted too little active substance, so it was decided to close that prolonged experiment. But within four weeks scheduled to complete the work, they managed to identify the structure of the primary active substance. It was N-acetyl-5-methoxytryptamine, which Lerner named “MT” (from Greek “melas”—black, and “tonin”—derived from serotonin).

Lerner presented his discovery to the public in a one-page article published in 1958 in the Journal of the American Chemical Society [23].

In the pineal gland, the MT production by pinealocytes is controlled by a circadian signal from the SCN, which is associated with the photoperiod. The activation of SCN neurons by ambient light perceived by the retina suppresses the synthesis of MT [12] (Figure 2).

MT, in turn, can markedly weaken the activity of SCN. This further stimulates MT secretion at night and contributes to an overall increase in the amplitude of circadian rhythms. As an endocrine messenger, MT quantitatively transfers the light signal to other tissues, expressing its own receptors and thereby delivering time-related information to the body [24].

The time-related and functional interaction between MT and SCN, and their reaction to ambient light ensure the synchronization of circadian rhythms of the body’s functional systems with regular changes in the environment. Hypothalamic paraventricular nuclei act as the most important relay between SCN and melatonin synthesis in the pineal gland, providing stable tonic stimulation of pineal gland activity at night, which is inhibited during the day by the release of GABA due to SCN. Pineal MT plays a crucial role in regulating seasonal rhythms as well as metabolism, immune response, reproductive function, and other vital physiological processes [25,26].

The immunomodulatory effect of MT is based on an endocrine response of circulating immunological cells and precursor cells in the bone marrow, which express its receptors. It has been shown that the rhythmic synthesis of MT is necessary for modulating both circadian and seasonal fluctuations of immune functions and for the effective functioning of the immune-pineal axis [27]. MT affects growth processes and thyroid gland hormone synthesis [28].

For the time being, MT is viewed as an integral regulator of biological rhythms and the most powerful endogenous antioxidant.

## 3. Extrapineal Melatonin

MT content in the body is determined not only by pineal gland secretion, whose contribution amounts to 5% [29], but also by the extrapineal localization of MT producing cells and intracellular sources of synthesis, changes in the volume of extracellular fluid, binding of hormones to blood proteins, metabolic and excretion rates depending on various external and internal regulatory factors. Cells that produce extrapineal MT are an integral part of the diffuse neuroimmuneendocrine system (DNIES) in which we distinguish central and peripheral population levels of MT-synthesizing cells [13].

The central level includes pineal cells and visual system cells, in which MT secretion depends on environment illumination [30], while in other tissues there is no such mechanism of MT production by DNIES endocrine cells. Furthermore, the MT synthesis (not only by endocrine cells but also by nerve, immune, and other cells) determines a uniquely wide range of its involvement in almost all vital physiological functions of the body.

While numerous hormones have been verified over the recent years outside “classical” locations of their formation, MT occupies an exclusive position with regard to the diversity of places where it is synthesized and secreted, which affords grounds for considering MT as the “cornerstone” of neuroimmunoendocrinology.

It was the discovery of MT synthesis in endocrine organs (pineal gland), neural structures (Purkinje cells in the cerebellum, retinal photoreceptors), and immunocompetent cells (T lymphocytes, NK cells, mast cells) that triggered the evolution of new approaches to the unified signal regulation of homeostasis, which, at the turn of 21st century, lead to the creation of a new integral biomedical discipline—neuroimmunoendocrinology [30].

### 3.1. Melatonin in the Nervous System

The evolutionarily formed capability of producing extrapineal MT in the brain and expressing its receptors determined the protection and restoration of life-sustaining activity under the impact of adverse factors [31]. The MT production in the brain is evidenced by the existence of its synthesis enzymes (AANAT and HIOMT) in various structures and mRNA coding them [32,33], and its metabolites N1-acetyl-N2-formyl-5-methoxykynuramine (AFMK) and N1-acetyl-5-methoxykynuramine (AMK) [34]. The latter are considered powerful antioxidants capable of absorbing free radicals and activating the cascade of free radical scavenging [35]. AANAT and HIOMT enzymes have been found in astrocytes, which can secret MT in cell cultures of the rat cortex and C6 glioma cell line [2,36]. The activity of AANAT was observed in the hippocampus, striate body, cerebellum, olfactory bulb, and prefrontal cortex [37,38]. MT production was found to be particularly high in certain brain regions, including the hypothalamus, pons cerebelli, medulla oblongata, and cerebellum [39,40]. It has been shown that MT production is not associated with the photoperiodic environment, while changes in MT subcellular distribution are due to specifics of their metabolic activity and the formation of toxic oxygen derivatives [9]. The increased MT synthesis and discharge in the brain in the case of acute hypoxia and traumatic brain injury in humans [41] are mediated by stress in which MT acts as a potent neuroprotector and antioxidant [42]. In cultivated neural stem cells from the subventricular zone of adult mice, researchers found a large amount of MT, and its production was even higher when these cells differentiated into mature neurons. The actual absence of glia cells in those cultures indicates that neurons themselves can synthesize MT. The synthesis and presence of MT in neurons and/or glia indicate that locally generated MT may impose intra-, para-, and autocrine functions in the brain, possibly regulating neural homeostasis by controlling the release of neurotransmitters and neurogenesis. Moreover, MT exerts an anti-excitotoxic and protective effect on the brain by decreasing the glutamate activity and increasing GABA [10]. It is shown that melatonin treatment after brain injury reduces astrocyte reactivity and neuronal cell apoptosis in the hippocampus and dentate gyrus [43].

Melatonin, a well-known anti-inflammatory drug, suppresses the levels of IL-6, TNF-α, and IL-1β in animal models of brain ischemia/reperfusion injury, subarachnoid hemorrhage, traumatic brain injury, and by inhibiting of reactive astrocytes also suppresses microglial activation, attenuating neuroinflammation in the brain [44,45].

Melatonin interacts in multiple ways with microglia, both directly and, via routes of crosstalk with astrocytes and neurons, indirectly. The immunological interplay in the CNS, with microglia playing a central role, is of high complexity and includes signaling toward endothelial cells and other leukocytes by cytokines. Melatonin interferes with these processes in multiple signaling routes and steps. In addition to canonical signal transduction by MT1 and MT2 melatonin receptors, secondary and tertiary signaling is of relevant and has to be considered, e.g., via the upregulation of sirtuins and the modulation of pro- and anti-inflammatory microRNAs [46].

It has been found that MT synthesized in the cerebellum interacts with its own receptors in the same region where it is synthesized and plays a protective role in the case of bacterial oxidative stress caused by lipopolysaccharides [47,48]. The anti-inflammatory and antioxidant properties of MT are important for the protection of neurons from damage during hypoxia or the impact of toxins and other adverse factors. Astrocytes, which amount to 70% of brain cells and perform multiple important functions in the central nervous system, also represent a local source of MT without circadian changes [49]. MT stimulates dendritogenesis [50], modulates the cytoskeleton organization in hippocampal cells [51], and promotes endogenous neurogenesis of the brain through MT2 receptors [52].

The retina is in a synthesis location where MT performs numerous regulatory functions [53]. The retinal pigment epithelium performs the local homeostatic function [54]. MT exerts receptor-mediated and non-mediated effects on the retina. MT receptors have been identified in multiple eye tissues, including retinal neuron and pigment epithelium, ciliary, cornea, sclera, and eye lens [55]. A lack of local MT synthesis in the retina plays a certain role in the pathogenesis of some eye diseases, including keratopathy, macular degeneration, and pathologies caused by ultraviolet radiation/light. MT can protect photoreceptor against photo-oxidative stress and counter ischemic cell damage [56]. Researchers have also noted the eye lens’s ability to synthesize MT [57].

### 3.2. Melatonin in the Immune System

The thymus, mast cells, natural killer cells, eosinophilic leukocytes, thrombocytes, and endothelial cells contain MT in various concentrations [58,59]. The MT synthesis in the thymus and compensatory growth of its production is observed in the absence of an epiphyseal source of indole [60]. Endogenous MT together with pineal MT and other hormonal and non-hormonal agents modulates and regulates the thymus function and homeostasis [61]. It increases the production of some thymic peptides [62] and prevents apoptosis in the thymus [63]. The existence of MT receptors in the thymus confirms this correlation [64]. Different localizations of MT in thymocyte subpopulations [65], each having definite production of cytokines and peptides [66], as well as a certain proliferation level, explains both the proliferative and antiproliferative properties of MT [67] in the thymus. The bone marrow is also a tissue with a substantially increased content of MT [2]. While pineal MT exerts an endocrine action reaching thymus target cells through the blood flow, extrapineal MT plays a key role as an intracrine, autocrine, and paracrine substance in the tissue where it is synthesized [68]. Melatonin restores neutrophil functions and prevents apoptosis amid a dysfunctional glutathione redox system [69]. It is known that in the process of immune response, activated leukocytes (macrophages, lymphocytes, neutrophils, mast cells) produce MT whose concentration is 100–1000 times higher than its level in the blood flow, thereby contributing to the improvement of bacterial phagocytosis and restoration of the damaged region [1,70,71]. Multiple actions of lymphocyte MT, including intra-, auto-, and paracrine regulation of IL-2 and/or IL-2R, have been confirmed by experimental data [58]. All these cell types express MT receptors which mediate some of its paracrine and autocrine effects. In most immunocompetent cells, MT or one of its metabolites exerts anti-inflammatory and antioxidant effects, providing a defensive mechanism that avoids tissue damage and the occurrence of chronic inflammatory diseases. In most immunocompetent cells, MT or one of its metabolites exerts anti-inflammatory and antioxidant effects, providing a defensive mechanism which allows to avoid tissue damage and occurrence of chronic inflammatory diseases. Melatonin has a powerful immunomodulatory activity [72]. It regulates the expression of cytokine genes, reducing the binding of NF-kB to DNA and thereby reducing the production of pro-inflammatory cytokines [73]. The anti-inflammatory effect of melatonin is also associated with an increase in the production of interleukin (IL)-4. By suppressing the expression of the inducible nitric oxide synthase (iNOS) gene, cyclooxygenase, as well as protein lipase A2, lipoxygenase and cytokine activity, melatonin prevents the development of systemic inflammation [74] Melatonin restores T-helper cell activity and interleukin-2 (IL-2) production, playing a critical role in regulating immune balance. The melatonin rhythm can influence the neuroendocrine system and modulate the natural immune reactivity, which together with the phagocytic system constitutes the first line of immunological defense. Melatonin participates in determining the threshold of sensitivity for specific immune activation. A robust melatonin rhythm might thus prevent infectious events together with malignant proliferation. This would result in a decreased frequency of specific and acute activation of the immune system, which is perceived as a stress event due to its neuroendocrine correlates, and documents the existence of an important physiological link between the neuroendocrine system and the immune system. [75].

Table 1 shows examples of target genes/signaling pathways that implement the effects of MT in a living organism.

### 3.3. Melatonin in the Gastrointestinal Tract

In the gastrointestinal tract, MT was first found in enterochromaffin cells of the human vermiform appendix [79]. Given the large total surface of the gastrointestinal tract and relatively high concentrations of MT per gram of tissue, it has been computed that the amount of produced MT is 400–500 times higher than in the pineal gland [80,81,82,83]. MT production is not circadian but governed by food intake [84,85]. MT effects within the gastrointestinal tract are mediated by membrane receptors MT1 and MT2. Besides membrane receptors, MT can also bind to some nuclear receptors, including RZR/RORγ [86]. For the time being, MT1 and MT2 receptors have been identified in the mitochondrial membrane of gastric endothelial cells. The researchers supposed that some physiological effects of MT on the gastrointestinal tract (e.g., angiogenesis) could be mediated by its mitochondrial, not membrane receptors [87]. As an amphiphilic molecule, MT penetrates biological membranes and reaches its targets inside and outside cells. The existence of specific transporters in cellular and mitochondrial membranes [88] facilitates MT transport against the gradient of its concentration. The presence of MT inside and outside cells and a variety of receptors determine its pleiotropic physiological and pharmacological effects in the gastrointestinal tract [89]. MT produced by enterochromaffin cells of the gastrointestinal mucosa is released into blood vessels or, through diffusion, reaches the outer layers of smooth muscles, where it acts as an antagonist of serotonin contractile effects, causing relaxation [90]. It regulates the transmembrane transport of electrolytes and ions, water content in the intestine, and mitotic activity [91].

In response to neuronal stimuli, proximal duodenum enterochromaffin cells discharge MT, which binds to MT2 receptors causing calcium release and bicarbonate ions secretion from adjacent lining cells, which neutralizes the acid content in the stomach as it is released into the duodenum [92,93]. MT plays an important role in the neurohumoral regulation of the duodenal mucosal barrier via a nicotinic receptor [94], as well as in the optimization of enzyme function in the gastrointestinal tract and microbiota content [95].

MT is synthesized in hepatocytes, where its subcellular distribution has proved to be higher than in the serum. Indole is metabolized in the liver and discharged with bile. Because of its antioxidant and anti-inflammatory properties, MT protects the gall bladder and intestinal epithelium induced by bile acids, oxysterol, or other products of metabolism during digestion. The intracellular MT content in the liver changes during the day and is not associated with changes in MT levels in the blood serum or food intake [9].

As in other organs, MT not only scavenges a wide range of reactive oxygen species but also optimizes the regulation of various antioxidant enzymes and performs the anti-inflammatory function in the gastrointestinal tract [58], including the suppression of prostaglandin synthesis and adhesion molecules, inhibition of leukocyte migration, and expression of cyclooxygenase-2 and anti-inflammatory cytokines [96]. MT can restore the gut microbiota balance, promoting the growth of bacteroides which are beneficial bacteria in the gastrointestinal tract [97]. Gastrointestinal MT, binding to membrane receptors and/or its intracellular and extracellular signaling molecules during the physiological activity, participates in the regulation of motility, hydrochloric acid production, cell proliferation, microbiota balance, prostaglandin synthesis, mediates mucosal immune cells, microbial metabolism, and rhythm crosstalk [15,98,99]. It protects gastrointestinal tissues against damages caused by oxidative stress and inflammation.

### 3.4. Melatonin in Other Visceral Organs

The presence and expression of the genes coding for arylalkylamine N-acetyltransferase (AANAT) and acetylserotonin methyl transferase (ASMT), the enzymes that control melatonin synthesis, have been identified in the heart. Local MT production has been observed in cardiomyocytes, where it is not photoperiodically controlled but changes during the day. It has been noted that the mitochondrial MT content in the heart over 24 h is significantly higher than in the brain, which could be evidence of a correlation between mitochondria activity and MT content. It has been argued that changes in MT levels in various organelles over 24 h could be due to its usage (e.g., as a radical scavenger) in combination with small fluctuations of synthesis, but not to changes in the synthesis alone [100]. The MT1 and MT2 receptors are present in human cardiomyocytes, arteries and left ventricles arteries [93]. Melatonin can reduce ROS generation, preserve mitochondrial stability, restore a robust mitochondrial function for unabated ATP production in cardiac tissues, protect against autophagy and apoptosis in cardiomyocytes via regulation of mitochondrial uncoupling protein 2 (UCP2). However, the protective mechanism of melatonin goes far beyond this action, involving different roles in the nervous system, myocardium remodeling and antioxidant properties; all factors generally involved in its receptor-mediated actions linked to antiarrhythmic properties [101]. Melatonin plays a pivotal role in the maintenance of calcium homeostasis in cardiomyocytes. Based on these mechanisms and their implications for the aging process, melatonin is considered a key antiaging element through several pathways [102]. Melatonin is known to play a special role in the functional development of the fetal cardiovascular system. In the early stages, it regulates the expression of clock genes (bmal1 and per2) in the fetal heart. By the time of birth it provides optimal functioning of the cardiovascular system by synchronizing the circadian oscillators in the heart, vessels, and their coordinating brain centers. This synchronizing effect of melatonin depends on the density of its receptors in various structures that control the volume and vascular resistance [103].

MT is synthesized in the skin, where it participates in the regulating of hair growth and pigmentation, and inhibits the proliferation of melanoma cells [104]. MT’s antioxidant activity protects against ultraviolet and X-ray radiation, burns, and other impacts that can cause skin damage. Much more MT is produced in the skin for its own use than can be found in the blood serum [105].

MT plays the key role in reproduction physiology by regulating the production of prolactin, follicle-stimulating and luteinizing hormones [1]. MT synthesis in the ovaries and testicles reflects auto- and paracrine regulation of reproductive physiology, guaranteeing high quality ovum and sperm [106,107]. Indole is produced in the cells of epithelium, stroma, myometrium and participates in maintaining the organ’s homeostasis by regulating multiple paths related to the processes of decidualization and implantation [108]. MT1 and MT2 receptors also play an important role in reproduction as MT regulates p38 activation, optimizing the viability of the embryo implant [109]. In women, MT synthesized by follicular cells participates in ovulation and placenta tissue homeostasis [110]. Extrapineal MT in the ovary provides local hormonal regulation and improves the potential for oocyte development thanks to its anti-inflammatory and antioxidant effects, mediated by a significant decrease in levels of inducible nitric oxide synthase and nitrogen oxide (NO) in luteinized cells and by an increase in mRNA levels for antioxidant transcription factor [111,112]. It has been demonstrated that MT concentration in the follicular preovulatory fluid is approximately three times higher than in the blood serum [113], similar to during pregnancy [114].

MT is also synthesized in the embryo, maintaining the normal status of DNA methylation by inducing the expression and synthesis of the TET2 protein, which transforms methylcytosine (the methylated base of DNA) into 5-hydroxymethylcytosine, and promotes embryo development [115].

The positive effect of MT, observed in most studies, on embryo preimplantation development and implantation could be explained by its antioxidant properties, which enable it to offset one of the factors most harmful for the oocytes and embryo, i.e. ROS hyperproduction. Furthermore, through MT1 and MT2 receptors located on the endometrium, oocyte, and blastocyst membranes, MT can activate signaling cascades, which leads to an increase in the synthesis of factors that promote implantation (LIF, p53) and inhibits factors that prevent it (MUC1) [116,117].

It has been shown that MT-generating enzymes (AANAT and ASMT) are produced throughout pregnancy, with an optimal expression in the third trimester [118]. Already in the first trimester of pregnancy, placental trophoblasts not only synthesize MT but also express its classical transmembrane receptors MT1 and MT2 [119,120], which points to the capability of locally synthesized MT to exert paracrine, autocrine, and/or intracrine effects in the placenta [121]. Cytotrophoblast and syncytiotrophoblast cells not only contain MT1 and MT2 membrane receptors but also synthesize MT [122]. MT produced in cytotrophoblast and syncytiotrophoblast cells, syncytial capillary membranes, syncytial kidneys, and knots in stromal cells and vascular endothelium is a direct scavenger of free radicals, stimulating the antioxidant enzyme activity and regulating the process of cytotrophoblast cells differentiation and apoptosis [118,123,124]. It preserves the balance of cytotrophoblast and syncytiotrophoblast cells, thus maintaining placental homeostasis [120].

The placenta as a part of the diffuse neuroimmunoendocrine system plays an extremely important role in the regulation of relationships between the mother and fetus, implementing the ontogenetic program of the fetus growth and development by producing classical peptide hormones, biogenic amines, messenger proteins, intra- and intercellular signal molecules [125,126]. As we know, it is MT and its circadian secretion rhythm that ensure successful placental development in health [122,127]. Thus, MT and its metabolites act as direct scavengers of free radicals formed during pregnancy, stimulating antioxidant enzymes, thereby providing stable protection against free radical damage at the cellular and tissue levels within the mother-placenta-fetus system [118,128,129]. Because of its ability to suppress the expression of inducible NO-synthase and cyclooxygenase genes, MT restricts the production of proinflammatory molecules (prostanoids, leukotrienes, cytokines, etc.), thereby providing anti-inflammatory protection [130]. As an immunomodulator and vascular-platelet hemostasis regulator, MT is involved in implantation, placentation, morphological and functional development of the placenta, and preservation of its neuroimmune-endocrine function aimed at the formation and engagement of vital functional systems of the fetus [24]. MT is also present in the amniotic fluid [131].

Placental MT passes into the maternal bloodstream in the third trimester at the latest, which substantially increases its level near the pregnancy term [108]. From the moment of fertilization and ovoimplantation, maternal MT participates in the mechanisms of regulating the hormone-producing function of the placenta and onset of its circadian rhythm, controlling gene expression (Bmal1, Perl3, Cry1-2, Clock, VEGE) [124]. Easily reaching the fetus, it plays the key role in its morphological and functional development and the formation of circadian rhythms [128,132,133].

The first endocrine cells appear in the fetus’s rectum and colon in the 6th–9th week of prenatal development [134]. Later, their number progressively increases, and the maximum frequency distribution is observed in the vermiform appendix during the intrauterine development of the fetus. This points to a significant role of the intestinal endocrine system, particularly the role of MT in regulating the mechanisms of embryo histogenesis and functional development of the gastrointestinal tract [135]. MT receptors are found in central and peripheral fetal tissues at the earliest stages of intrauterine development [136,137]. Maternal MT synchronizes peripheral circadian oscillators in these organs and coordinates their function with clock gene rhythms of suprachiasmatic nuclei and other tissues, including the adenohypophysis and adrenal gland [5,138].

During antenatal life, the fetal suprachiasmatic nuclei and organs are circadian oscillators whose rhythmic activity is triggered by and depends on the state of circadian organization of vital activities in the mother and its primary messenger biorhythms generated by suprachiasmatic nuclei, i.e., MT. This ensures the integration of endogenous biorhythms of the baby’s functional systems into the adult-like circadian system, which is regulated by its suprachiasmatic nuclei in dependence to circadian changes in environment illumination [139]. Further maturation of the central rhythm driver continues after the child’s birth, and MT produced by mononuclear cells, polymorphonuclear phagocytes, endothelial cells in the mammary glands and passed with the milk also contributes to supporting and developing of clock genes in brain cortex neurons and in other regions of the central nervous system [140].

Thus, in prenatal ontogenesis extrapineal MT is the key molecule that directs and coordinates the genetic process of the fetus’s morphological and functional development, which is crucial for successful postnatal adaptation to a new environment and a healthy life in subsequent months and years (Figure 3).

## 4. Melatonin and Mitochondria

Mitochondria, which are present in all cells of the body (except erythrocytes), play the key role in their metabolism, calcium homeostasis, apoptosis, and regulation of multiple physiological and pathological processes [141]. They are responsible for the production of energy (ATP) which results from glucose metabolism (glycolysis) and cellular respiration (oxidative phosphorylation) in the inner mitochondrial membrane. Glycolysis in the cell’s cytosol generates pyruvate, which is actively transported to the mitochondrial matrix where it is metabolized to acetyl-coenzyme A (acetyl-CoA) under the action of pyruvate dehydrogenase complex (PDC) enzyme. Acetyl-CoA makes an important contribution to the tricarbonic acid cycle and thereby couples glycolysis to ATP production [142,143]. Acetyl-CoA is also an important co-factor of arylalkylamine N-acetyltransferase (AANAT) which converts serotonin to N-acetylserotonin—a MT precursor [144]. Being responsible for energy supply to cells by means of oxidative phosphorylation, mitochondria are the main location for generating reactive oxygen species (ROS) – oxidative phosphorylation by-products that participate in the regulation of cellular oxidation-reduction homeostasis. The presence of nitric-oxide synthase (mtNOS) in mitochondria points to the production of nitric oxide (NO) and peroxynitrite (ONOO^−^), which are produced in these organelles in case of inflammation [145]. In addition to ATP, mitochondria also produce precursors to synthesis macromolecules such as DNA/RNA, proteins, and lipids [73]. Furthermore, they participate in maintaining cellular Ca2^+^ homeostasis [146]. Moreover, mitochondria regulate cellular apoptosis by releasing apoptotic factors (cytochrome C) and activating caspases [10,141] (Figure 4).

Counteraction to oxidative stress and reprogramming of impaired metabolism in cells is provided by MT synthesized in mitochondria where its distribution is much higher than in other subcellular organelles and does not have a circadian rhythm [9,76]. It has been found that MT is released from mitochondria and then, through the MT1 receptor on the membrane, controls the discharge of cytochrome C, i.e., provides automitocrine regulation. It has been shown that the density of MT1 and MT2 receptors is very high on the mitochondrial membranes of gastric endothelial cells [82].

MT is present in mitochondrial membranes and passes into mitochondria with the help of oligopeptide transporters PEPT1 and PEPT2 located on the membrane [147]. The existence of MT in mitochondria as a result of its uptake and de novo synthesis provides certain functional benefits to these organelles and cells at large [148]. High concentrations of MT and its multiple actions as an antioxidant provide powerful protection to these organelles exposed to the impact of free radicals [130]. Inside mitochondria, MT acts as a direct scavenger of free radicals and related non-radical products, and stimulates antioxidant enzymes, including superoxide dismutase 2 (SOD2), catalase, and glutathione reductase, while suppressing pro-oxidant enzymes [149,150,151,152]. Each of these enzymes participates in maintaining the oxidation-reduction homeostasis of mitochondria, acting as good scavengers of reactive oxygen and nitrogen species [2,153,154,155].

MT can suppress nitric-oxide synthase which is responsible for nitric oxide formation, and lipoxygenase promoting the formation of anion superoxide [156]. By regulating lipoxygenase activity, MT protects cells against hydroperoxidation of polyunsaturated fatty acids [157]. It modulates reactions of the endoplasmic reticulum to stress [158], sirtuin activity [159], mitophagy and autophagy processes [160].

MT exerts a significant effect on the mitochondrial function by virtue of its capability to increase the efficiency of the electron transport chain and enhance ATP production [106]. Such improvement of the mitochondrial function and ATP synthesis increases the electron transport rate and reduces ROS production, avoiding a harmful decrease in the mitochondrial membrane potential. Under certain circumstances, MT reduces oxygen consumption by mitochondria, which could protect these organelles against excessive oxidative stress and prevent oxidative degradation of mitochondrial DNA [155]. Loss of such a potent mitochondrial antioxidant as MT may have serious consequences in terms of oxidative stress enhancement and reduced ATP production followed by increased formation of free radicals [130,161]. These changes have dire consequences for mitochondrial and cellular physiology and are common in mitochondrial diseases [162].

Thus, MT, through direct scavenging of ROS in mitochondria, antioxidant protection activation and membrane integrity preservation, plays a crucial role in maintaining the normal mitochondrial functions and energy exchange in cells. Further research into MT’s effects on mitochondrial functions in various tissues and organs in health and disease, as well as studies into MT medications targeted at mitochondria, will help us get insight into the mechanisms of its protective action and its use for the treatment of numerous pathologies.

## 5. Melatonin as a Neuroimmunoendocrine Marker and Molecular Target for Socially Significant Diseases

Studies into the mechanisms controlling the vital activities at the subcellular, cellular, and tissue levels, and the interactions of functional systems under various environmental conditions have determined the pivotal role of MT production decrease, mostly pineal MT, in the processes of aging, tumor growth, age-related degenerative diseases, and such pathological states as obesity, diabetes mellitus, metabolic syndrome, hypertension, gastrointestinal tract disorders, etc. [73]. The main factor promoting their development is oxidative stress that causes damage to cellular macromolecules (including DNA, proteins) and lipids, as a result of mitochondrial dysfunction due to MT depletion in those organelles. Disruption of antioxidant processes of electron transfer inside the cell and oxidative phosphorylation, peroxidation of mitochondrial membrane phospholipids, changes in the membrane fluidity and permeability—all these leads to mitochondrial dysfunction with aging and can be the primary reason for a wide range of pathological conditions [4].

Numerous studies in the fields of neurology, endocrinology, cardiology, oncology, reproductive and perinatal medicine have demonstrated the key role of MT in the protection of tissues against damage through beneficial effects on oxidative phosphorylation of mitochondria, antioxidant, and free radical activity, through immunomodulatory and anti-apoptotic effects, as well as maintenance of chronobiological homeostasis [103,163,164,165,166,167]. MT treatment of brain ischemia increases survivability and reduces neurodegeneration and postischemic neuron loss, providing evidence for MTs’ potential as a neuron protector in both adults and children [152,168].

Because of MT’s involvement in the regulating of inflammatory processes, carbon and lipid metabolism, its application is effective for prophylactic treatment of atherosclerosis, ischemic brain injury, cerebrovascular diseases, etc. [169,170,171]. MT had a beneficial effect on reducing ROS production and protecting of pancreatic beta cells in patients with diabetes mellitus [172]. It has been shown that MT exerts antitumor effects associated with mitochondrial function regulation. MT provides a fundamental protective mechanism because of its ability to change metabolism in cancer cells, which causes increased apoptosis, thereby inhibiting their growth in lung cancer, mammary gland, and colon tumors, and their metastasis [173].

Aging is a primary risk factor for neurodegenerative diseases (cognitive impairment of moderate severity, Alzheimer’s disease, Parkinson’s disease, and other forms of dementia progressing in connection with low MT secretion) [173,174,175]. In experimental models of Alzheimer’s and Parkinson’s disease, the observed neurodegeneration was partially prevented by MT [176]. Clinical studies have also proved the efficacy of MT in improving cognitive disorders and decreasing specific symptoms such as disorganized thinking, excitement, emotional lability, and attention deficit [177]. The neurodegeneration process in Parkinson’s disease is associated with neuroinflammation, microglia activation, and lymphocytic infiltration. By regulating the activity of antioxidant enzymes, mitochondrial complex-I and glutathione (GSH), MT reduced oxidative stress in the experimental model of Parkinson’s disease [178], and MT administration to patients slowed down neurodegeneration and improved the quality of sleep [179,180].

## 6. Conclusions

In conclusion, MT is involved in maintaining homeostasis in various organ tissues and protecting their functional activity under exposure to unfavorable environmental conditions.

A wide range of functions carried out by extrapineal MT, its endocrine, paracrine, and autocrine actions to protect tissues against damage are determined by its synthesis in mitochondria and regulation of oxidation-reduction homeostasis in cells. The realization of the antioxidant and anti-inflammatory properties of MT requires its high concentrations in tissues in comparison with its level in the blood serum. Disruption of these processes leads to mitochondrial dysfunction and the development of various pathophysiological conditions (cancer, inflammation, fibrosis, Alzheimer’s disease, etc.).

The results of experimental and clinical studies indicate the importance of timely diagnosis of violations of the pineal and extrapineal MT production, which can act as a neuroimmunoendocrine marker of various diseases [6].

In addition, given the fact that MT is widely distributed in various organs and tissues and is involved in the molecular mechanisms of various pathological processes, it can also be considered as a possible target in the selection of personalized therapy to optimize the treatment of socially significant diseases.

Further research on MT can also contribute to the developing of new approaches to its use for the prevention and treatment of age-related pathology.

## Figures and Tables

**Figure 1 ijms-23-01835-f001:**
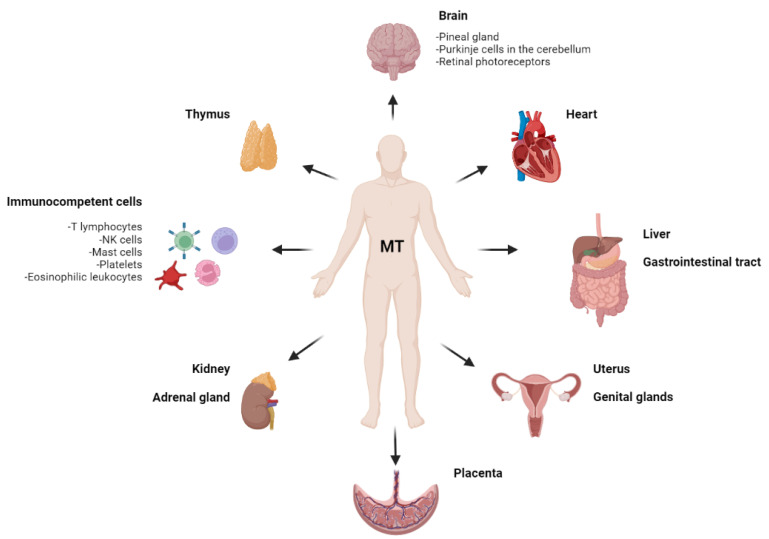
Melatonin localization in the human body. The figure illustrates the unique property of melatonin to have the most widespread localization in the human body, being synthesized in various organs.

**Figure 2 ijms-23-01835-f002:**
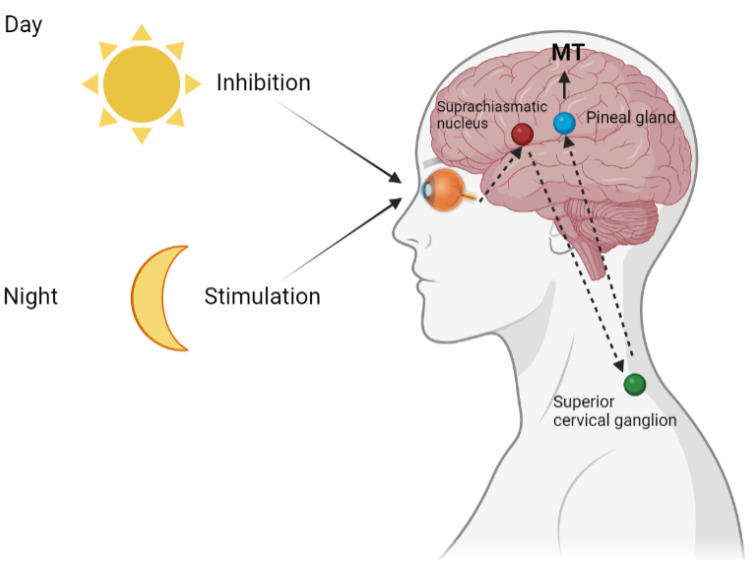
Melatonin is the main regulator of the circadian rhythm in all living organisms. Light information from the rods and cones of the retina through the ganglion cells and directly from the light-sensitive ganglion cells enters the paired suprachiasmatic nucleus (SCN) of the hypothalamus. These signals then travel to the cervical spinal cord, from where they travel back to the brain and reach the pineal gland. During sleep in the dark, when most of the SCN neurons are inactive, the nerve endings release norepinephrine, which activates the synthesis of melatonin in the pinealocytes. Bright light blocks the synthesis, while in constant darkness, the rhythmic production, maintained by the periodic activity of the SCN, is preserved.

**Figure 3 ijms-23-01835-f003:**
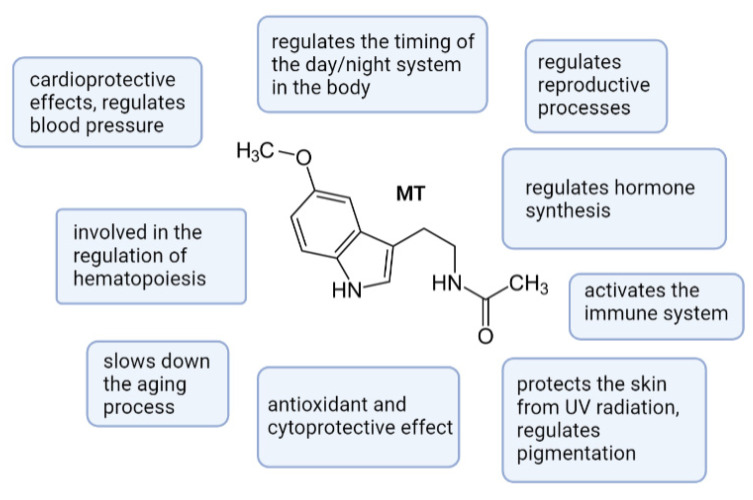
Diagram of the biological role of melatonin.

**Figure 4 ijms-23-01835-f004:**
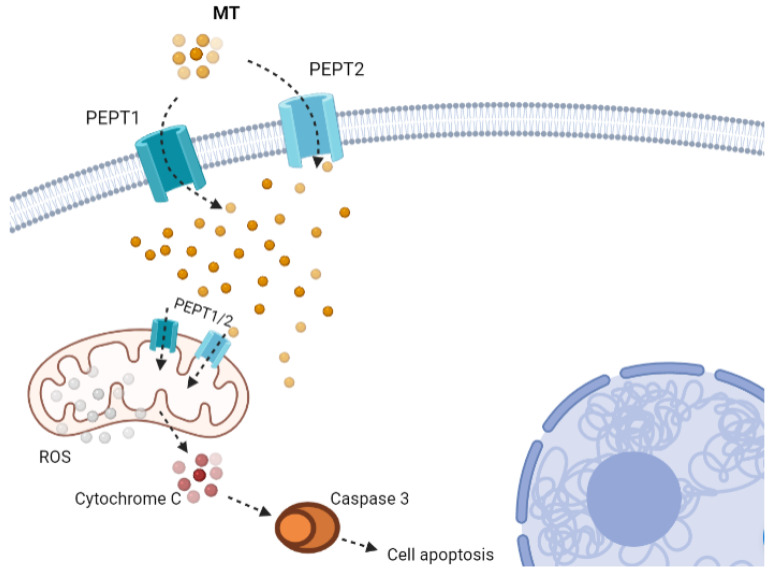
Melatonin and mitochondrial function. The high concentration of melatonin in mitochondria is due to the fact that PEPT1/2 proteins present in the mitochondrial membrane move melatonin into mitochondria against the gradient. Since melatonin is a powerful free radical scavenger, its presence in mitochondria reflects the participation of this hormone in compensatory reactions that occur during the development of mitochondrial aging and pathology associated with mitochondrial dysfunction ([133], modified).

**Table 1 ijms-23-01835-t001:** Examples of key genes and signaling pathways mediating the effects of MT.

Gene-Targets/Signaling Pathways	MT Effects	Physiological/Pathological Manifestations
PTEN/AKT [76]	Stimulating	Anti-inflammation
SIRT1 [77]	Stimulating	Anti-aging
TOLLR4 [77]	Stimulating	Anti-aging, immunomodulation
iNO [77]	Inhibitory	Anti-oxidative
NRF-2, CBR1, CLPP, SOD2 [78]	Inhibitory	Anti-oxidative
ANAPC4, HSPA4/Ubiquitination pathway [77]	Inhibitory	Neuroprotection

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
