# Peer review of "Melatonin as the Cornerstone of Neuroimmunoendocrinology"

_ijms, 2022, doi:10.3390/ijms23031835_

Round 1
Reviewer 1 Report
The authors reviewed major data regarding the role of melatonin in various physiological and pathological processes, and showing that neuroimmunoendocrine system is strongly dependent on melatonin regulatory action of body homeostasis. The paper is very interesting and well- organized. However, some criticisms need to be addressed.
Major points
- A third type of MT receptor, MT3, also called quinone reductase 2, is found in the cytosol, amd plays an important role in the modulation of redox homeostasis. Please mention this receptor and its role in the Introduction.
- About the role of melatonin in nervous system, two very recent papers (doi: 10.32598/bcn.12.2.986.1; doi: 10.3390/cells10102647suggest that melatonin treatment after brain injury reduces astrocyte reactivity, as well as neuronal cell apoptosis and neuroinflammation. Other papers on the protective/reparative action of melatonin in brain ischemia/reperfusion models are worthy too be mentioned beside the role of MT in excitoxic damage.
- Given the focus of the review on neuroendocrinoimmunology one would expect a long chapter on melatonin functions in immune cells, instead of the short paragraph Melatonin in the immune system. Most importantly, the references cited in this paragraph only range from 1995 to 2013. However, several papers have been published in the last eight years. Therefore, an implementation of information about this topic should be performed in the present review.
- Given the interaction of melatonin with nuclear receptors, it would be useful listing in a Table gene targets of melatonin-regulated gene transcription, in particular those most relevant for the antioxidant and immunomodulatory action of melatonin.
- In general, the reference list should be updated by adding other more recent papers, and more recent findings, published in the last four years, should be reported in the paragraphs 3.3., 3.4 and chapter 4.
Minor points
Several typos and some grammar errors need to be corrected.
Few examples:
Lines 55-56 rephrase the sentence, some words are missing
Line 57 Check and correct contradictory definitions of melatonin “hydrophilic”, “lipophilic”
Line 136 replace “unifield” with “unified”
Line 364 Replace “aralkylamine” with “arylalkylamine”
Line 375 Replace caspase with caspases
Line 394 add “formation” after “responsible for nitric oxide”
Author Response
Major points
- A third type of MT receptor, MT3, also called quinone reductase 2, is found in the cytosol, amd plays an important role in the modulation of redox homeostasis. Please mention this receptor and its role in the Introduction.
comments are taken into account and recommendations are implemented
- About the role of melatonin in nervous system, two very recent papers (doi: 10.32598/bcn.12.2.986.1; doi: 10.3390/cells10102647suggest that melatonin treatment after brain injury reduces astrocyte reactivity, as well as neuronal cell apoptosis and neuroinflammation. Other papers on the protective/reparative action of melatonin in brain ischemia/reperfusion models are worthy too be mentioned beside the role of MT in excitoxic damage.
comments are taken into account and recommendations are implemented
- Given the focus of the review on neuroendocrinoimmunology one would expect a long chapter on melatonin functions in immune cells, instead of the short paragraph Melatonin in the immune system. Most importantly, the references cited in this paragraph only range from 1995 to 2013. However, several papers have been published in the last eight years. Therefore, an implementation of information about this topic should be performed in the present review.
comments are taken into account and recommendations are implemented
- Given the interaction of melatonin with nuclear receptors, it would be useful listing in a Table gene targets of melatonin-regulated gene transcription, in particular those most relevant for the antioxidant and immunomodulatory action of melatonin.
comments are taken into account and recommendations are implemented
- In general, the reference list should be updated by adding other more recent papers, and more recent findings, published in the last four years, should be reported in the paragraphs 3.3., 3.4 and chapter 4.
comments are taken into account and recommendations are implemented
Minor points
Several typos and some grammar errors need to be corrected.
Few examples:
Lines 55-56 rephrase the sentence, some words are missing
Line 57 Check and correct contradictory definitions of melatonin “hydrophilic”, “lipophilic”
Line 136 replace “unifield” with “unified”
Line 364 Replace “aralkylamine” with “arylalkylamine”
Line 375 Replace caspase with caspases
Line 394 add “formation” after “responsible for nitric oxide”
These and some other errors are fixed
Reviewer 2 Report
The authors provided a comprehensive review in an interesting topic. However, I think authors can add some new information. I suggest adding the effects of melatonin on pro-oxidant enzymes such as NADPH Oxidases, iNOS, COX-2, etc. As melatonin can suppress the expression and activity of these enzymes, authors may propose these mechanisms as protective mechanisms of melatonin in the brain.
Author Response
The authors provided a comprehensive review in an interesting topic. However, I think authors can add some new information. I suggest adding the effects of melatonin on pro-oxidant enzymes such as NADPH Oxidases, iNOS, COX-2, etc. As melatonin can suppress the expression and activity of these enzymes, authors may propose these mechanisms as protective mechanisms of melatonin in the brain.
Recommended information included in the text
Reviewer 3 Report
The Authors in the abstract wrote: "This review provides an overview and discussion of the major data regarding the role of melatonin in various physiological and pathological processes, which affords grounds for considering melatonin as the "cornerstone"on which neuroimmunoendocrinology has been built as an integral concept of homeostasis regulation”. Unfortunately, I do not see any "integral concept" in the article showing that melatonin is indeed the "cornerstone" of neuroimmunoendocrinology. Although the Authors have reviewed many papers, the submitted manuscript appears to be a cluster of random and fairly general conclusions. There is no in-depth analysis of the literature, there are plenty of general statements like "melatonin regulates this or that". The article does not give a clear summary of the knowledge gathered so far in the field chosen by the Authors. The whole concept of "neuroedocrinology" does not exist in the manuscript at all. The data is, in my opinion, a random patchwork of some very superficial information about melatonin and how it works in the brain, cells of the immune system, and other tissues. However, it is hard to find an actual discussion of how melatonin integrates cells of the endocrine, nervous and immune systems. Thus, in my opinion the Authors did not achieve the goal they set for themselves. In general, the idea of the article is very unclear, there is no keynote. The sections on „Mitochondria” and „Socially Important Diseases” do not seem to be directly related to the preceding paragraphs. Thus, I recommend to reject the article.
Author Response
The Authors in the abstract wrote: "This review provides an overview and discussion of the major data regarding the role of melatonin in various physiological and pathological processes, which affords grounds for considering melatonin as the "cornerstone"on which neuroimmunoendocrinology has been built as an integral concept of homeostasis regulation”. Unfortunately, I do not see any "integral concept" in the article showing that melatonin is indeed the "cornerstone" of neuroimmunoendocrinology.
In the Extrapineal Melatonin section, lines 124-140, we explain why melatonin can be considered the cornerstone of a new integral science - neuroimmunoendocrinology, namely, we write: Cells that produce extrapineal MT are an integral part of the diffuse neuroimmuneendocrine system (DNIES) in which we distinguish central and peripheral population levels of MT-synthesizing cells [29]. The central level includes pineal cells and visual system cells, in which MT secretion depends on environment illumination [30], while in other tissues there is no such mechanism of MT production by DNIES endocrine cells. Furthermore, the MT synthesis (not only by endocrine cells but also by nerve, immune, and other cells) determines a uniquely wide range of its involvement in almost all vital physiological functions of the body. While numerous hormones have been verified over the recent years outside “classical” locations of their formation, MT occupies an exclusive position with regard to diversity of places where it is synthesized and secreted, which affords grounds for considering MT as the “cornerstone” of neuroimmunoendocrinology.It was the discovery of MT synthesis in endocrine organs (pineal gland), neural structures (Purkinje cells in the cerebellum, retinal photoreceptors), and immunocompetent cells (T lymphocytes, NK cells, mast cells) that triggered the evolution of new approaches to the unified signal regulation of homeostasis, which, at the turn of 21st century, lead to the creation of a new integral biomedical discipline — neuroimmunoendocrinology [30].
The sections on „Mitochondria” and „Socially Important Diseases” do not seem to be directly related to the preceding paragraphs.
It seems to us that these sections just reflect the importance of melatonin as an integral hormone of the diffuse neuroimmunoendocrine system, which, having many general regulatory properties, including being a powerful scavenger of free radicals, plays an important role as a marker and target in the development of a number of socially significant diseases that can be attributed to the so-called "Mitochondrial diseases" (neurodegenerative processes, etc.).
Thus, I recommend to reject the article.
Of course, this is the right of the reviewer to recommend an article to be accepted or rejected. We hope that our explanations will be taken into account in the decision of the editorial board, as well as the opinions of three other reviewers, who evaluate the article as a whole rather positively and offered constructive comments, which we introduced into the revised text.
Reviewer 4 Report
Many roles of melatonin(MT) in diffuse neuroimmunoendocrine system(DNIES) as well as diverse synthesis sites in different tissues are presented. However, the manuscript lacks clarity with numerous, non-coherent descriptions on the role of MT in different physiological processes. Moreover, it needs more clear argument for the use of the word such as cornerstone. Overall, the manuscript has only one Figure and is not well organized. The reviewer finds much room for improvement as follows.
- Figure 1. needs more description in legend.
- In pineal MT section, a Figure on circadian rhythm can be added.
- line 226: its -> it
- A table or Figure summarizing the biological roles of MT must be added.
- A Figure in section 4 (Melatonin and mitochondria) would help the reader.
- A Figure in section 5 would help the reader.
- Line 461: 'Thus,' may better be removed.
Author Response
We have added a description for figure 1. Here it is:
The figure illustrates the unique property of melatonin to have the most widespread localization in the human body, being synthesized in various organs
We made 3 more drawings and inserted them into the corresponding sections.
Round 2
Reviewer 1 Report
The authors satisfactorily addressed all raised criticisms. After revision the manuscript quality was significantly improved, and the paper is worthy to be published.
Author Response
The authors satisfactorily addressed all raised criticisms. After revision the manuscript quality was significantly improved, and the paper is worthy to be published.
Thank you very much.
Reviewer 3 Report
I appreciate the changes made by the Authors to the manuscript, but despite their efforts, I maintain my first assessment and still perceive the article as any cluster of numerous and interesting but random facts that do not constitute a review article, so I still recommend that the submitted manuscript should be rejected.
Author Response
I appreciate the changes made by the Authors to the manuscript, but despite their efforts, I maintain my first assessment and still perceive the article as any cluster of numerous and interesting but random facts that do not constitute a review article, so I still recommend that the submitted manuscript should be rejected.
We are very sorry.
Reviewer 4 Report
The revised manuscript now seems suitable for publication.
Author Response
The revised manuscript now seems suitable for publication.
Thank you very much.